# The Impact of KRAS Status on the Required Surgical Margin Width for Colorectal Liver Metastasis Resection

**DOI:** 10.3390/jcm12062313

**Published:** 2023-03-16

**Authors:** Kentaro Iwaki, Satoshi Kaihara, Tatsuya Koyama, Kai Nakao, Shotaro Matsuda, Kan Toriguchi, Koji Kitamura, Nobu Oshima, Masato Kondo, Hiroki Hashida, Hiroyuki Kobayashi, Kenji Uryuhara

**Affiliations:** Department of Surgery, Kobe City Medical Center General Hospital, 2-1-1 Minatojimaminamimachi, Chuo-ku, Kobe 650-0046, Japan

**Keywords:** colorectal metastasis, liver resection, local recurrence, surgical margin, KRAS, wild type, mutant

## Abstract

Local recurrence after colorectal liver metastasis (CRLM) resection severely affects survival; however, the required surgical margin width remains controversial. This study investigated the impact of KRAS status on surgical margin width and local recurrence rate (LRR) post-CRLM resection. Overall, 146 resected CRLMs with KRAS status (wild-type KRAS (wtKRAS): 98, KRAS mutant (mKRAS): 48) were included. The LRR for each group, R1 (margin positive) and R0 (margin negative), was analyzed by KRAS status. R0 was further stratified into Ra (margin ≥ 5 mm) and Rb (margin < 5 mm). Patients with local recurrence had significantly worse 5-year overall survival than those without local recurrence (*p* = 0.0036). The mKRAS LRR was significantly higher than wtKRAS LRR (*p* = 0.0145). R1 resection resulted in significantly higher LRRs than R0 resection for both wtKRAS and mKRAS (*p* = 0.0068 and *p* = 0.0204, respectively), and while no significant difference was observed in the Ra and Rb LRR with wtKRAS, the Rb LRR with mKRAS (33.3%) was significantly higher than Ra LRR (5.9%) (*p* = 0.0289). Thus, R0 resection is sufficient for CRLM with wtKRAS; however, CRLM with mKRAS requires resection with a margin of at least 5 mm to prevent local recurrence.

## 1. Introduction

Globally, colorectal cancer is the third leading cause of cancer death, and the most common site of colorectal cancer metastasis is the liver [1,2]. Recently, the advances in multidisciplinary treatment have improved the prognosis of stage IV colorectal cancer with liver metastases; however, hepatectomy is still the best hope for colorectal liver metastasis (CRLM) [3,4].

The margin status of CRLM resection and local recurrence are closely related, and insufficient surgical margins result in local recurrence, which is a strong prognostic factor [5,6,7,8]. Our group previously reported that local recurrence after CRLM resection resulted in a worse prognosis than that associated with other recurrences [8]. Hence, the local control of the liver should be achieved to ensure an appropriate surgical margin width; however, the appropriate surgical margin width required remains controversial. The “1 cm rule” was traditionally advocated [9]. Recently, Kokudo et al. have reported “2 mm” as the minimum surgical margin [10]. Some studies have reported similar outcomes in patients with margins < 10 mm to those of patients with a 10 mm margin [11,12]. Differences in recurrence patterns based on surgical margin width and the impact of chemotherapy have also been reported [12,13].

Moreover, gene diagnosis has recently gained attention, and KRAS status is important in colorectal cancer treatment as a surrogate marker of tumor malignancy. Mutations in the RAS family genes correlate with the efficacy of anti-epidermal growth factor receptor antibodies [14]. KRAS mutations (mKRAS) have been reported to be associated with poor survival outcomes after CRLM resection and a higher risk of local recurrence and lung metastasis [15,16]. The different histopathological morphologies based on the presence or absence of RAS mutations have been reported to affect overall survival (OS). Furthermore, these differences may affect the required margin width [17]. However, the association between KRAS mutations and the surgical margin width of CRLM resection for local control remains unclear. Hatta et al. reported that the resection margin status in patients with wild-type KRAS (wtKRAS) was more important with respect to OS than with mKRAS [18]. Thus, we hypothesized that mKRAS tumors, generally associated with higher malignancy, may require a larger resection margin width for local control. No studies have used local recurrence as an endpoint when considering the surgical margin width for achieving ideal liver resection.

Therefore, in this study, we investigated the impact of KRAS status on surgical margin width for local recurrence rates (LRR) after CRLM resection.

## 2. Materials and Methods

### 2.1. Ethical Considerations

All patients provided informed consent before inclusion in the study. This study was conducted in accordance with the Declaration of Helsinki and approved by our institutional review board (approval number: Zn230207).

### 2.2. Patients and Study Design 

This was a single-center, retrospective study. The inclusion criteria were as follows: CRLM with KRAS status, complete macroscopic resection, description of surgical margins, and the absence of other cancers. The exclusion criteria were as follows: R2 resections, unresectable extrahepatic metastases, and BRAF mutations. BRAF mutations, while having a severe prognostic impact, are extremely rare and different in nature from RAS mutations. Therefore, they were excluded from this study. According to our previous study, CRLMs in contact with the Glissonean pedicle were also excluded because of their high LRR [8]. Based on these criteria, 146 CRLM resected from 67 patients between 2009 and 2019 at the Kobe City Medical Center General Hospital were included in this study. 

First, the surgical margin width was classified into two categories: R1 (margin positive) and R0 (margin negative). Moreover, R0 was grouped as Ra (surgical margin ≥ 5 mm) or Rb (surgical margin < 5 mm). The cutoff value of the surgical margin width was determined using receiver operating characteristic (ROC) curve analysis. The primary endpoint was the LRR in each group, and LRRs were compared for each surgical margin width (R1, R0, Ra, and Rb).

### 2.3. KRAS Mutation Analysis

Prior to 2015, we measured codons 12 and 13. Codons 12, 13, 59, 61, 117, and 146 were measured using the MEBGEN RASKET™-B Kit (Medical & Biological Laboratories Co., Ltd., Tokyo, Japan) in accordance with our revised national guidelines, from 2015 to present. In this study, all measured KRAS mutations were considered positive [19,20,21].

### 2.4. Preoperative Management and Surgical Procedure

Preoperative contrast-enhanced computed tomography (CT), magnetic resonance imaging (MRI), and intraoperative ultrasonography were routinely performed. The Indocyanine green plasma clearance rate and 99 m Tc-galactosyl human serum albumin single-photon emission CT were used to evaluate the preoperative remnant liver reserve in major hepatectomy cases, as described in our previous study [22].

We used a Cavitron ultrasonic surgical aspirator (CUSA) (Integra Life Sciences, Princeton, NJ, USA) for liver parenchymal dissection, and a VIO soft-coagulation system (Erbe Elektromedizin GmbH, Tübingen, Germany) for hemostasis. Our standard operative procedure was parenchymal-sparing hepatectomy for R0 as described in our previous study [8]. Anatomical resection was performed for tumors in contact with a Glissonean pedicle.

### 2.5. Pathological Examination

The surgical margin width was determined to be the smallest distance between the tumor and the resection surface. The remaining macroscopic tumor was defined as R2 resection. Exposure of the tumor on the resection surface or microscopic infiltration within the margin was defined as R1 resection. The absence of microscopic infiltration within the surface was defined as R0 resection.

### 2.6. Follow-Up

Patients were followed-up every three months, and CT was performed every 6 months to investigate recurrence until 5 years after surgery. Postoperative chemotherapy was administered on a case-by-case basis. The median observation period was 1227 days (interquartile range (IQR): 639–1854 days).

### 2.7. Definition of Recurrence Type

If recurrence was detected, it was categorized as either local recurrence, intrahepatic recurrence excluding local recurrence, extrahepatic recurrence, or peritoneal dissemination. Each resection site was precisely identified by the preoperative CT, MRI, surgical findings, and postoperative CT, which also revealed whether local recurrences were present. Local recurrence was defined as recurrence on the surface of remaining liver tissue. Intrahepatic recurrence was defined as hepatic recurrence excluding local recurrence. 

### 2.8. Statistical Analyses

The JMP Pro13 software was used for statistical analyses (SAS Institute, Cary, NC, USA). Continuous data were expressed as medians and IQRs. ROC curve analysis was used to determine the cutoff values for the univariate studies. Categorical data were analyzed using the chi-square test, and continuous data were analyzed using Student’s *t*-test. Kaplan–Meier plots were used to analyze the 5-year OS and log-rank tests were used to compare the groups. Statistical significance was set as *p*-value < 0.05.

## 3. Results

Table 1 presents the patients and CRLMs characteristics. Overall, 146 tumors resected from 67 patients were analyzed in this study. Based on KRAS, 98 tumors were wtKRAS and 48 were mKRAS. The median tumor size and surgical margin width were 15 mm (IQR 10–27 mm) and 3 mm (IQR 1–5 mm), respectively. Anatomical resection was performed for 24 tumors.

Local recurrence was observed in 26 tumors (17.8%). To determine the cutoff values for surgical margin width, ROC curve analysis was performed (Figure 1). 

To determine the cutoff values for surgical margin width, a receiver operating characteristic (ROC) curve analysis was performed, and 5 mm was determined as cutoff value of surgical margin.

The cutoff value surgical margin was determined to be 5 mm. R0 was divided into Ra (margin ≥ 5 mm) and Rb (margin < 5 mm) after the analysis of LRR of R1 and R0. The total number of CRLMs in each group was 28, 118, 48, and 70 for R1, R0, Ra, and Rb, respectively. Table 2 presents the result of the univariate analysis for wtKRAS and mKRAS tumors. There were no significant differences in the operative procedure, tumor size, or surgical margin width between the two groups.

Figure 2 presents Kaplan–Meier plots of the presence or absence of local recurrence in total (Figure 2a), wtKRAS (Figure 2b), and mKRAS (Figure 2c) groups. Five-year OS of patients with local recurrence and without local recurrence was 34.0% and 65.8%, respectively, in total. The local recurrence group had a significantly worse 5-year OS than the non-local recurrence group in total (*p* = 0.0036). For mKRAS, the 5-year OS of the local recurrence group (26.6%) was also significantly worse than that of the non-local recurrence group (61.3%) (*p* = 0.0319).

Overall, the local recurrence group had significantly worse 5-year OS than non-local recurrence group (*p* = 0.0036). 5 year-OS of the local recurrence group was also significantly worse than that of the non-local recurrence group for mKRAS (*p* = 0.0319).

Overall survival (OS), KRAS mutations (mKRAS), wild-type KRAS (wtKRAS).

The sites of recurrence of each group are presented in Table 3. The LRRs of mKRAS and wtKRAS tumors were 29.2% and 12.2%, respectively. The LRR of patients with mKRAS was significantly higher than that of patients with wild-type KRAS tumors (*p* = 0.0145). No significant differences were observed in intrahepatic recurrence, extrahepatic recurrence, and peritoneal dissemination between the two groups.

Table 4 presents the LRRs for each surgical margin in wtKRAS and mKRAS tumors. The LLR of R1 resection was significantly higher than that of R0 resection in the entire cohort (42.9% vs. 11.9%, *p* = 0.0004); however, no significant difference was observed in the LRR of Ra and Rb (8.3% vs. 14.3%, *p* = 0.3168). In wtKRAS, the LRR of R1 was significantly higher than that of R0 (33.3% vs. 7.5%, *p* = 0.0068). The LRR of Ra and Rb in wtKRAS were 9.7% and 6.1%, respectively, which did not show statistically difference. In mKRAS, the LRR of R1 and R0 was 60.0% and 21.1%, respectively. The LRR of R1 was also significantly higher than that of R0 (*p* = 0.0204). Moreover, the LRR of Ra and Rb was 5.9% and 33.3%, respectively. The LRR of Rb was significantly higher than that of Ra in the mKRAS group (*p* = 0.0289).

We further stratified R0 resected CRLMs using 1 mm and 10 mm cutoffs and analyzed their LRRs for the Appendix A (Table A1). There was no significant difference in LRR in the entire cohort, either at a cutoff of 1 mm or 10 mm. No significant difference in LRRs was observed in wtKRAS when the cutoff was 1 mm or 10 mm. In mKRAS, LRRs of margin ≥ 1 mm and margin < 1 mm were 10.3% and 50.0%, this difference was statistically different (*p* = 0.0189); however, no significant difference was observed in LRR with a margin of 10 mm or more compared with a margin of 10 mm or less.

## 4. Discussion

Hepatectomy is the most promising treatment for CRLM. In this study, local recurrence after hepatectomy had a negative impact on OS, similar to previous studies [5,6,7,8]. The goal of CRLM surgery is local control of the liver to cure colorectal cancer. Therefore, we need to clarify the appropriate surgical margin width to achieve it. 

Several studies have assessed the required surgical margin width. Although a few studies have reported that the era of advances in chemotherapy and multimodal therapies may make R1 resection acceptable [12], R0 resection for CRLM is the gold standard in the world [3,23,24] and the “1 cm rule” has been widely adopted. A surgical margin of > 1 cm offers a low LRR and favorable prognosis in some studies [5,9,25]; however, sometimes a 1 cm resection margin is difficult to achieve due to a shortage of the remnant liver reserve and the location of the major intrahepatic vessels. Recently, the noninferiority of a smaller surgical margin width has been reported [10,26,27]. The LRR of a surgical margin ≥ 1 mm was not significantly different from that of a surgical margin < 1 mm in the R0 resection cohort [8]. Ausania reported an acceptable LLR of surgical margins < 1 mm compared to a 0 mm margin [24]. LRRs of R1 resection were higher than those of R0 in this study, although a larger surgical margin width did not show superiority in the LRR of wtKRAS. In these findings, R0 resection is necessary, and the required surgical margin, ranging from 1–10 mm, may depend on the tumor malignancy. Our previous study reported that tumors in contact with the Gllisonean pedicle should be resected because of a high LRR in case of detachment from the Gllisonean pedicle [8]. The locations of the tumor and intrahepatic vessels also play a role in determination of the width of the required surgical margin width.

The KRAS status is one of the surrogate markers for tumor biology. A higher risk of local recurrence and worse prognosis after mKRAS CRLM resection have been reported [15,16]. However, the relationship between the KRAS status and resection margins remains unknown. Hatta et al. reported that resection margin status is more important in patients with wtKRAS than in those with mKRAS [18]. In this previous report, the primary endpoints were OS and recurrence-free survival, which were affected by chemotherapy, making it difficult to assess the impact of liver resection and surgical margin width. Hence, we focused on the local recurrence after hepatectomy in the present study. This is the first study to analyze the impact of KRAS status on the required surgical margin width for CRLM resection.

The LRR of mKRAS was higher than that of wtKRAS tumors, which is in accordance with previous studies reporting a high recurrence rate and poor OS in patients with mKRAS tumors. Additionally, more surgical margin width was needed for local control of patients with mKRAS tumors based on our result. Our hypothesis that mKRAS tumors associated with higher malignancy may require larger resection margin widths for local control is confirmed. Generally, mKRAS had a higher alignancy than wtKRAS tumors [15,18,28]. This could be due to various reasons. First, mKRAS was strongly associated with a poor response to chemotherapy in patients with CRLM [29]. Although no strong evidence has been reported for the efficacy of preoperative chemotherapy for CRLM, many experts highlight its importance in reducing local and early postoperative recurrences. Furthermore, it is speculated that mKRAS tumors may be resistant to chemotherapy, resulting in residual live tumor cells at the margins, contributing to local recurrence. More than half of the patients received preoperative chemotherapy in the present study and the response was neither radiologically nor pathologically investigated at this time. Thus, further research is needed. Second, the tumor border configuration (TBC) may be relevant. Jass et al. first described TBC as marginal findings [30]. They defined infiltrating TBC as tumors exhibiting an invasive border or an unrecognizable surgical margin between the tumor and normal tissue. Pushing TBC indicated that tumors exhibited well-circumscribed borders between the tumor and normal tissue. Studies on tumor marginal findings, budding, and TBC are ongoing, and the prognosis for infiltrative TBC is poor in colorectal cancer [31,32,33]. Fonseca et al. reported the presence of TBC in CRLM [17]. Poorly differentiated clusters, defined as clusters comprising five or more tumor cells within a tumor without glandular structures, were worse prognostic factors than the pseudo capsule type. Furthermore, surgical margins ≥ 10 mm did not affect OS. Although no studies have characterized KRAS and TBC in CRLM, the high LRR of mKRAS may be associated with a worse TBC, infiltration type, or poorly differentiated clusters. Additional research from a pathological perspective is needed in the future.

KRAS status was an important factor in determining the surgical procedure in our study. CRLM with mKRAS requires a surgical margin of at least 5 mm but not more than 10 mm. There may be cases in which attempting to secure a 5 mm surgical margin would result in an insufficient margin and require a small anatomical resection. If the case is metachronous CRLM, examination of the previously operated primary tumor will reveal the KRAS status. Hence, if an mKRAS tumor is detected, a surgical margin width of at least 5 mm should be planned. However, for CRLM with wtKRAS, a surgical margin of 1 mm is sufficient for local control, and the surgery may be less difficult and minimally invasive. Instead of one treatment/surgical option for all, pretreatment predictors for cancer, such as liver cancer and patients’ genetic makeup, should be considered for cancer treatment for improved patient health and economy outcomes. These considerations should be made pre-surgically, with more emphasis on patients’ epi/genetic makeup [34,35,36].

This study had several limitations. This was a single-center, retrospective study. The sample size was small; particularly, a limited number of samples had wide surgical margin widths. Patients with tumors of varying origins were included in this study. Therefore, large prospective studies are warranted. The aim of this study was to investigate the impact of KRAS status on the surgical margin width for LRR, and the primary endpoint was the LRR. To observe local recurrence, the tumor and not the patient, should be targeted individually [37]. If the number of tumors per patient is skewed between KRAS-mutation-positive and -negative groups, the study results may be biased. The number of tumors per patient in both groups was not significantly different between the two groups in this study. We would like to consider limiting the observation to single liver metastases. The relationship between preoperative and postoperative chemotherapy and the resection margin was not analyzed in this study and is a topic for future research. Studies from a pathological perspective, such as TBC, may also lead to new findings.

## 5. Conclusions

R0 resection is sufficient for CRLM with wtKRAS; however, CRLM with mKRAS requires resection with a surgical margin width of at least 5 mm to prevent local recurrence.

## Figures and Tables

**Figure 1 jcm-12-02313-f001:**
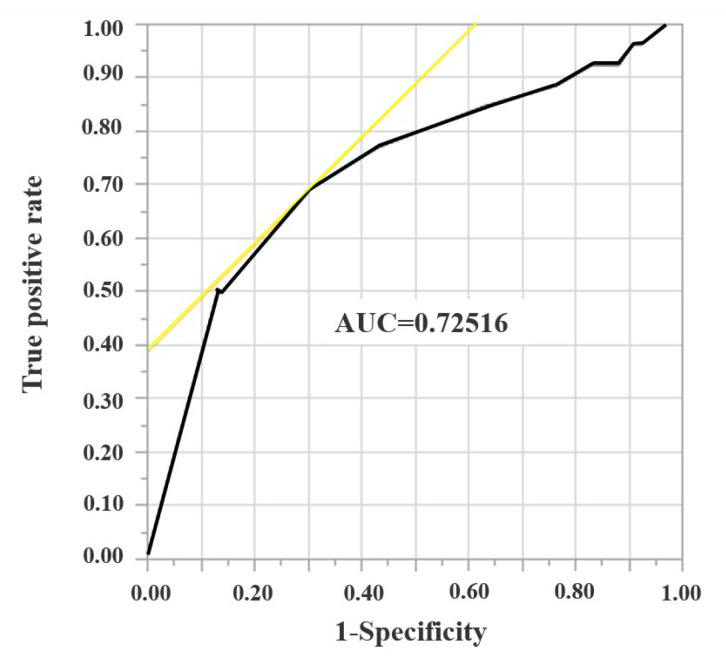
A receiver operating characteristic curve analysis for surgical margin width.

**Figure 2 jcm-12-02313-f002:**
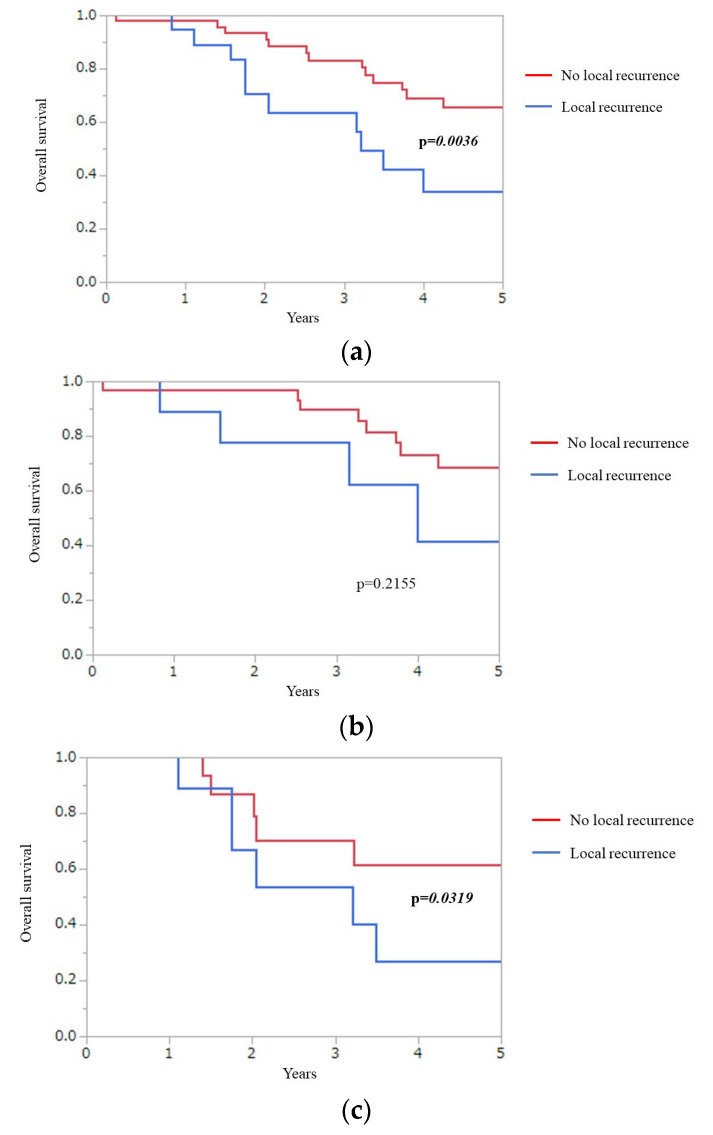
Kaplan–Meier plots of presence or absence of local recurrence group in total (**a**), wtKRAS (**b**), and mKRAS (**c**).

**Table 1 jcm-12-02313-t001:** CRLM and patients’ characteristics.

Patient	n = 67
Age (years) (median, IQR)	69, 63–74
Synchronous/metachronous tumor	38/29
Number of tumors (median, IQR)	1, 1–2
Preoperative chemotherapy	42
Postoperative chemotherapy	47
Colon/Rectum	45/22
CEA (ng/mL) (median, IQR)	7.9, 3.1–17.7
CRLM	n = 146
wtKRAS/mKRAS	98/48
Anatomical resection	24
Tumor size (mm) (median, IQR)	15, 10–27
Surgical margin (mm) (median, IQR)	3, 1–5
Local recurrence	26

Continuous data are expressed as the medians and interquartile ranges. Carcinoembryonic antigen (CEA), colorectal liver metastasis (CRLM), interquartile ranges (IQR), KRAS mutant (mKRAS), wild-type KRAS (wtKRAS).

**Table 2 jcm-12-02313-t002:** Univariate analysis of wtKRAS and mKRAS tumor group.

	wtKRAS (n = 98)	mKRAS (n = 48)	*p*-Value
Anatomical resection	16	8	0.9585
Tumor size (mm) (median, IQR)	15, 10–27	16, 10–30	0.7573
Surgical margin (mm) (median, IQR)	3, 1–6	2, 1–5	0.0747
R1	18	10	0.7236
R0	80	38	
Ra (margin ≥ 5 mm)	31	17	0.5362
Rb (margin < 5 mm)	49	21	

The LRR of mKRAS was significantly higher than that of wtKRAS (*p* = 0.0145). Interquartile ranges (IQR), KRAS mutant (mKRAS), wild-type KRAS (wtKRAS), positive surgical margin (R1), negative surgical margin (R0), surgical margin ≥ 5 mm (Ra), 0 < surgical margin < 5 mm (Rb).

**Table 3 jcm-12-02313-t003:** Recurrence site in each group.

	wtKRAS	mKRAS	*p*-Value
Local recurrence	12, 12.2%	14, 29.2%	0.0145
Intrahepatic recurrence (except local recurrence)	47, 48.0%	25, 52.1%	0.6396
Extrahepatic recurrence	33, 33.7%	21, 43.8%	0.2361
Peritoneal dissemination	11, 11.2%	6, 12.5%	0.8223

The LRRs of mKRAS and wtKRAS tumors were 29.2% and 12.2%, respectively. The LRR of patients with mKRAS was significantly higher than that of patients with wild-type KRAS tumors (*p* = 0.0145). KRAS mutant (mKRAS), wild-type KRAS (wtKRAS).

**Table 4 jcm-12-02313-t004:** Local recurrence rates for each surgical margin in wtKRAS and mKRAS.

	n	Local Recurrence	*p*-Value
Total	146		
R1	28	12, 42.9%	0.0004
R0	118	14, 11.9%	
Ra (margin ≥ 5 mm)	48	4, 8.3%	0.3168
Rb (margin < 5 mm)	70	10, 14.3%	
wtKRAS	98		
R1	18	6, 33.3%	0.0068
R0	80	6, 7.5%	
Ra (margin ≥ 5 mm)	31	3, 9.7%	0.5611
Rb (margin < 5 mm)	49	3, 6.1%	
mKRAS	48		
R1	10	6, 60.0%	0.0204
R0	38	8, 21.1%	
Ra (margin ≥ 5 mm)	17	1, 5.9%	0.0289
Rb (margin < 5 mm)	21	7, 33.3%	

In wtKRAS tumors, the LRR of R1 was significantly higher than that of R0 (*p* = 0.0068). The LRR for Ra and Rb did not differ significantly. In mKRAS tumors, the LRR of R1 was also significantly higher than that of R0 (*p* = 0.0204). Moreover, the LRR of Ra and Rb was 5.9% and 33.3%, respectively. The LRR of Rb was significantly higher than that of Ra in the mKRAS group (*p* = 0.0289). KRAS mutant (mKRAS), wild-type KRAS (wtKRAS), positive surgical margin (R1), negative surgical margin (R0), surgical margin ≥ 5 mm (Ra), 0 <surgical margin < 5 mm (Rb).

## Data Availability

The data presented in this study are available on request from the corresponding author. The data are not publicly available due to privacy and ethical.

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
