# Peer review of "The Impact of KRAS Status on the Required Surgical Margin Width for Colorectal Liver Metastasis Resection"

_jcm, 2023, doi:10.3390/jcm12062313_

Round 1

Reviewer 1 Report

This research article by Iwaki et al. 2023, titled “The Impact of KRAS Status on the Required Surgical Margin Width for Colorectal Liver Metastasis Resection  

Title in 16 words reflects the hypothesis, and the disease, patients, and looking for whats?.

The Abstract/Keywords Section

·                Abstract appropriately summarize the manuscript, with appropriate keywords,

·                But, need more key words, wild type and mutant to be added

·                No discrepancies between the abstract and the manuscript remainder,

·                The Abstract can be understood without reading the manuscript; however, the manuscript provides more clarifications and details.

The Introduction Section

·                The introduction is showing details,

·                The purpose of the study is not clearly defined, as missing to mention the computational analysis,

·                Authors provided a rationale for performing the study, but still not appropriate,

·                Authors defined terms used in the remainder of the manuscript,

·                Aim needs more clarification through definite objectives.

Methods and indexes need more elaboration and references

The Results Section

·                Results are clearly explained,

·                Results are reasonable,

Tables and Figures are appropriate, adequately showing results and are appropriately labeled, and legend provides a clear explanation, in the results.

Discussion concise,

·                Hypothesis was proposed, but the authors didn’t state whether the hypothesis was verified or falsified in the discussion part,  

·                Authors' conclusion(s) were justified by the results found in the study,

·                Authors didn’t note the “strength(s)” of the study,

·                Authors should mention the following “recommendation”

Instead of one treatment/surgical option-for-all, more importantly, pretreatment predictors for cancer as liver cancer [add these citations: Rizk HH, Hamdy NM, Al-Ansari NL, El-Mesallamy HO (2016) Pretreatment Predictors ofResponse to PegIFN-RBV Therapy in EgyptianPatients with HCV Genotype 4. PLoS ONE 11(4):e0153895

And

MO El-Derany, NM Hamdy, NL Al-Ansari, HO El-Mesallamy, Integrative role of vitamin D related and Interleukin-28B genes polymorphism in predicting treatment outcomes of Chronic Hepatitis C, BMC gastroenterology 16 (1), 1-12.] as well as patients genetic-makeup consideration for cancer treatment outcome [add this citation: Kamal AM, NM Hamdy, Hegab HM, El-Mesallamy HO. Expression of thioredoxin-1 (TXN) and its relation with oxidative DNA damage and treatment outcome in adult AML and ALL: A comparative study. Hematology. 2016.http://dx.doi.org/10.1080/10245332.2016.1173341.] will have better outcome for patients’ health and economy. This to be considered pre-surgically as well with more emphasis on patients’ epi/genetics’ make-up.

·                What are the “future prospective(s)” for completion of the work?

·                List of abbreviations are needed.

The References Section

·                More new relevant references are necessary,

The final recommendation Reconsider with Minor Revision.

Author Response

  March 9th, 2023

Editor-in-Chief

Dr. Emmanuel Andrès 
Special Issue Editor

Dr. Hiroki Hashida

Journal of Clinical Medicine

Subject: Resubmission of manuscript titled, “The Impact of KRAS Status on the Required Surgical Margin Width for Colorectal Liver Metastasis Resection.

Manuscript ID number for first submission: jcm-2239402.

Dear Editors and Reviewers:

Thank you very much for considering our revised manuscript now titled, “The Impact of KRAS Status on the Required Surgical Margin Width for Colorectal Liver Metastasis Resection.” by Iwaki et al, which was originally submitted to Journal of Clinical Medicine in February of this year.

In accordance with the Editor’s and Reviewers’ suggestions, we have answered your comments and modified our manuscript using bold font and underlined text.

Modifications and replies to reviewers’ comments are specified below.

We believe that we have made our best attempts to revise our paper in accordance with the comments of the reviewers and hope that our revised manuscript is now acceptable for publication.

We are looking forward to hearing from you at your earliest convenience.

Sincerely Yours,

Kentaro Iwaki, Satoshi Kaihara, Tatsuya Koyama, Kai Nakao, Shotaro Matsuda, Kan Toriguchi, Koji Kitamura, Nobu Oshima, Masato Kondo, Hiroki Hashida, Hiroyuki Kobayashi, Kenji Uryuhara.

Department of Surgery, Kobe City Medical Center General Hospital, 2-1-1 Minatojimaminamimachi, Chuo-ku, Kobe 650-0046, Japan.

Comments

Reviewer #1-1: Need more key words, wild type and mutant to be added.

Reply 1-1:

We appreciated your suggestion. We added the key words which you suggested.

Keywords: Colorectal metastasis; Liver resection; Local recurrence; Surgical margin, KRAS, Wild type, Mutant. on line 23.

Reviewer #1-2: The purpose of the study is not clearly defined, as missing to mention the computational analysis. Authors provided a rationale for performing the study, but still not appropriate. Aim needs more clarification through definite objectives.

Reply 1-2:

                 The purpose of the study is to investigate the impact of KRAS status on surgical margin width for local recurrence rates (LRR) after CRLM resection, as noted at the end of the Introduction section. We have provided two reasons for this, as noted in the Introduction section. First, the margin status and local recurrence after CRLM resection are strong prognostic factors; however, the appropriate surgical margin width required has been controversial. Second, KRAS mutations and the surgical margin width of CRLM resection for local control is still not clear. We revised the introduction section to emphasize these points as below:

The margin status of CRLM resection and local recurrence are closely related, and insufficient surgical margins result in local recurrence, which is a strong prognostic factor 5–8. Our group previously reported that local recurrence after CRLM resection resulted in a worse prognosis than that associated with other recurrences8. Hence, local control of the liver should be achieved to ensure an appropriate surgical margin width; however, the appropriate surgical margin width required remains controversial.

Furthermore, these differences may affect the required margin width18. However, the association between KRAS mutations and the surgical margin width of CRLM resection for local control remains unclear. Hatta et al. reported that the resection margin status in patients with wild-type KRAS (wtKRAS) was more important with respect to OS than with mKRAS19. Thus, we hypothesized that mKRAS tumors, generally associated with higher malignancy, may require a larger resection margin width for local control. No studies have used local recurrence as an endpoint when considering the surgical margin width for achieving ideal liver resection.

Reviewer #1-3:  Methods and indexes need more elaboration and references.

Reply 1-3:

                  We added this sentence on line 69. BRAF mutations, while having a severe prognostic impact, are extremely rare and different in nature from RAS mutations. Therefore, they were excluded from this study.

We revised with new references on line 82. Prior to 2015, we measured codons 12 and 13. Codons 12, 13, 59, 61, 117, and 146 were measured using the MEBGEN RASKET™-B Kit (Medical & Biological La-boratories Co., Ltd., Tokyo, Japan) in accordance with our revised national guidelines from 2015 to present. In this study, all measured KRAS mutations were considered positive20-22.

And we added this sentence on line 97. Anatomical resection was performed for tumors in contact with a Glissonean pedicle.

Reviewer #1-4: the authors didn’t state whether the hypothesis was verified or falsified in the discussion part, Authors' conclusion(s) were justified by the results found in the study, Authors didn’t note the “strength(s)” of the study, Authors should mention the following “recommendation.” Instead of one treatment/surgical option-for-all, more importantly, pretreatment predictors for cancer as liver cancer [add these citations: Rizk HH, Hamdy NM, Al-Ansari NL, El-Mesallamy HO (2016) Pretreatment Predictors ofResponse to PegIFN-RBV Therapy in EgyptianPatients with HCV Genotype 4. PLoS ONE 11(4):e0153895

And MO El-Derany, NM Hamdy, NL Al-Ansari, HO El-Mesallamy, Integrative role of vitamin D related and Interleukin-28B genes polymorphism in predicting treatment outcomes of Chronic Hepatitis C, BMC gastroenterology 16 (1), 1-12.] as well as patients genetic-makeup consideration for cancer treatment outcome [add this citation: Kamal AM, NM Hamdy, Hegab HM, El-Mesallamy HO. Expression of thioredoxin-1 (TXN) and its relation with oxidative DNA damage and treatment outcome in adult AML and ALL: A comparative study. Hematology. 2016.http://dx.doi.org/10.1080/10245332.2016.1173341.] will have better outcome for patients’ health and economy. This to be considered pre-surgically as well with more emphasis on patients’ epi/genetics’ make-up.

Reply 1-4:

Our hypothesis which mentioned introduction section was correct. We added in following sentence on line 241. Our hypothesis that mKRAS tumors associated with a higher malignancy may require larger resection margin widths for local control is confirmed.

We mentioned our strength of this study in the discussion section as several limitations. We have again reviewed and revised the limitations as follow. This study had several limitations. This was a single-center, retrospective study. The sample size was small; particularly, a limited number of samples had wide surgical margin widths. Patients with tumors of varying origins were included in this study. Therefore, large prospective studies are warranted. The aim of this study was to investigate the impact of KRAS status on the surgical margin width for LRR, and the primary endpoint was the LRR. To observe local recurrence, the tumor and not the patient, should be targeted individually38. If the number of tumors per patient is skewed between the KRAS mutation positive and negative groups, the study results may be biased. The number of tumors per patient in both groups was not significantly different between the two groups in this study. We would like to consider limiting the observation to single liver metastases. The relationship between preoperative and postoperative chemotherapy and the resection margin was not analyzed in this study and is a topic for future research. Studies from a pathological perspective, such as TBC, may also lead to new findings.

We still believe that our findings suggest an important fact.

Thank you for your suggestions. We added your recommended sentence and references on line 273.

Reviewer #1-5: What are the “future prospective(s)” for completion of the work?

Reply 1-5:

Our study had several limitations. this was a single-center, retrospective study. Secondly, the sample size was small. Therefore, large prospective studies are warranted. And as mentioned the last part of discussion section, further research is needed in terms of pathological findings and the effects of chemotherapy. We are looking at tumor border configuration. We plan to discuss resection margins from a more pathological standpoint. In this study, more than half of the patients received preoperative chemotherapy, and the response was neither radiologically nor pathologically investigated at this time. Thus, further research is needed as mentioned on discussion section. We also focused on the effect of preoperative chemotherapy and local recurrence control. We are now considering our next research method.

Reviewer #1-6: List of abbreviations are needed.

Reply 1-6:

Thank you for your comment. We added abbreviation section as below on line 303.

List of Abbreviations: Cavitron Ultrasonic Surgical Aspirator (CUSA), colorectal liver metastasis (CRLM), computed tomography (CT), interquartile range (IQR), KRAS mutation (mKRAS), local recurrence rate (LRR), magnetic resonance imaging (MRI), overall survival (OS), positive surgical margin (R1), negative surgical margin (R0), receiver operating characteristic (ROC), surgical margin ≥5 mm (Ra), 0<surgical margin<5 mm (Rb), tumor borders configuration (TBC), wildtype KRAS (wtKRAS).

Reviewer #1-7: More new relevant references are necessary.

Reply 1-7:

We cited more new relevant references as below. Thank you.

  1. Hashiguchi, Y.; Muro, K.; Saito, Y.; Ito, Y.; Ajioka, Y.; Hamaguchi, T.; Hasegawa, K.; Hotta, K.; Ishida, H.; Ishiguro, M.; et al. Japanese Society for Cancer of the Colon and Rectum (JSCCR) guidelines 2019 for the treatment of colorectal cancer. Int J Clin Oncol 2020, 25, 1-42. https://doi.org/10.1007/s10147-019-01485-z.
  2. Yoshino, T.; Muro, K.; Yamaguchi, K.; Nishina, T.; Denda, T.; Kudo, T.; Okamoto, W.; Taniguchi, H.; Akagi, K.; Kajiwara, T.; et al. Clinical validation of a multiplex kit for RAS Mutations in colorectal cancer: Results of the RASKET (RAS KEy Testing) Prospective, Multicenter Study. EBioMedicine 2015, 2, 317-23. https://doi.org/10.1016/j.ebiom.2015.02.007.
  3. Taniguchi, H.; Okamoto, W.; Muro, K.; Akagi, K.; Hara, H.; Nishina, T.; Kajiwara, T.; Denda, T.; Hironaka, S.; Kudo, T.; et al. Clinical validation of newly developed multiplex kit using Luminex xMAP technology for detecting simultaneous RAS and BRAF mutations in colorectal cancer: Results of the RASKET-B Study. Neoplasia 2018, 20, 1219-1226. https://doi.org/10.1016/j.neo.2018.10.004.
  4. Rizk, H.H.; Hamdy, N.M.; Al-Ansari, N.L.; El-Mesallamy, H.O. Pretreatment predictors of response to PegIFN-RBV therapy in Egyptian patients with HCV genotype 4. PLoS ONE, 2016, 11, e0153895. https://doi.org/10.1371/journal.pone.0153895.
  5. El-Derany, M.O.; Hamdy, N.M.; Al-Ansari, N.L.; El-Mesallamy, H.O. Integrative role of vitamin D related and interleukin-28B genes polymorphism in predicting treatment outcomes of chronic hepatitis C. BMC Gastroenterol 2016, 16, 19. http://doi.org/10.1186/s12876-016-0440-5.
  6. Kamal, A.M.; El-Hefny, N.H.; Hegab, H.M.; El-Mesallamy, H.O. Expression of thioredoxin-1 (TXN) and its relation with oxi-dative DNA damage and treatment outcome in adult AML and ALL: A comparative study. Hematology 2016, 21, 567-575. http://doi.org/10.1080/10245332.2016.1173341.
  7. Procopio, F.; Viganò, L.; Cimino, M.; Donadon, M.; Del Fabbro, D.; Torzilli, G. Does KRAS mutation status impact the risk of local recurrence after R1 vascular resection for colorectal liver metastasis? An observational cohort study. Eur J Surg Oncol 2020, 46, 818-824. http://doi.org/10.1016/j.ejso.2019.12.004.

Reviewer 2 Report

I wish to thank you for giving the opportunity to review this interesting manuscript.

The study presents a cohort of 67 patients who underwent 146 CRLM resection. The study assesses KRAS status, and it is relationship to required surgical margin.

I have several remarks:

1.       I would recommend explaining the reason for excluding patients with BRAF mutations (Material and methods section).

2.       I would recommend explaining why other KRAS mutations were not included, such as codon 61.  Why were other RAS family mutations not assessed? Did the authors assess KRAS mutation in the primary colon tumor also?

3.       In the results section line 119, the patient’s number is 67 not 46.

4.       As KRAS mutations most probably are found in all CRLM performed in the same patient, I would recommend the authors to complete analysis comparing both groups of 67 patients with wKRAS and mKRAS without separating CLRM for each patient.

5.       I would recommend showing all three graphs in figure 2 in the same page.

6.       Adjuvant treatment might affect recurrence and was not addressed/discussed in the results section.

7.       I would recommend elaborating more on limitations (see comment #2, #4 and #6). In addition, examination of previously operated primary tumor might affect the surgical margin (line 259-261) which might result in bias.

Author Response

  March 9th, 2023

Editor-in-Chief

Dr. Emmanuel Andrès 
Special Issue Editor

Dr. Hiroki Hashida

Journal of Clinical Medicine

Subject: Resubmission of manuscript titled, “The Impact of KRAS Status on the Required Surgical Margin Width for Colorectal Liver Metastasis Resection.

Manuscript ID number for first submission: jcm-2239402.

Dear Editors and Reviewers:

Thank you very much for considering our revised manuscript now titled, “The Impact of KRAS Status on the Required Surgical Margin Width for Colorectal Liver Metastasis Resection.” by Iwaki et al, which was originally submitted to Journal of Clinical Medicine in February of this year.

In accordance with the Editor’s and Reviewers’ suggestions, we have answered your comments and modified our manuscript using bold font and underlined text.

Modifications and replies to reviewers’ comments are specified below.

We believe that we have made our best attempts to revise our paper in accordance with the comments of the reviewers and hope that our revised manuscript is now acceptable for publication.

We are looking forward to hearing from you at your earliest convenience.

Sincerely Yours,

Kentaro Iwaki, Satoshi Kaihara, Tatsuya Koyama, Kai Nakao, Shotaro Matsuda, Kan Toriguchi, Koji Kitamura, Nobu Oshima, Masato Kondo, Hiroki Hashida, Hiroyuki Kobayashi, Kenji Uryuhara.

Department of Surgery, Kobe City Medical Center General Hospital, 2-1-1 Minatojimaminamimachi, Chuo-ku, Kobe 650-0046, Japan.

Reviewer #2-1: I would recommend explaining the reason for excluding patients with BRAF mutations (Material and methods section).

Reply 2-1:

We appreciate your comment. BRAF mutation is very rare. Hence, only encountered one BRAF mutation case, which was excluded from our cohort. BRAF mutation had a severe effect on prognosis. We would also like to investigate the effect of BRAF mutations on resection margins through a multicenter study. We added the following sentence on line 69. BRAF mutations, while having a severe prognostic impact, are extremely rare and different in nature from RAS mutations. Therefore, they were excluded from this study.

Reviewer #2-2:  I would recommend explaining why other KRAS mutations were not included, such as codon 61. Why were other RAS family mutations not assessed? Did the authors assess KRAS mutation in the primary colon tumor also?

Reply 2-2:

Thank you for your comment. As KRAS and NRAS mutations have been elucidated, Japanese guidelines have changed. KRAS testing at our hospitals has also changed. To give you more detail, prior to 2015, we only measured codons 12 and 13, but since 2015, we have measured codons 12, 13, 59, 61, 117, and 146 using MEBGEN RASKET™-B Kit (Medical & Biological Laboratories Co., Ltd., Tokyo, Japan). Based on this information, we revised “KRAS Mutation Analysis” section as follow. Prior to 2015, we measured codons 12 and 13. Codons 12, 13, 59, 61, 117, and 146 were measured using the MEBGEN RASKET™-B Kit (Medical & Biological La-boratories Co., Ltd., Tokyo, Japan) in accordance with our revised national guidelines, from 2015 to present. In this study, all measured KRAS mutations were considered positive. We added three references associated with above.

・Hashiguchi, Y.; Muro, K.; Saito, Y.; Ito, Y.; Ajioka, Y.; Hamaguchi, T.; Hasegawa, K.; Hotta, K.; Ishida, H.; Ishiguro, M.; et al. Japanese Society for Cancer of the Colon and Rectum (JSCCR) guidelines 2019 for the treatment of colorectal cancer. Int J Clin Oncol 2020, 25, 1-42. doi: 10.1007/s10147-019-01485-z.

・Yoshino, T.; Muro, K.; Yamaguchi, K.; Nishina, T.; Denda, T.; Kudo, T.; Okamoto, W.; Taniguchi, H.; Akagi, K.; Kajiwara, T.; et al. Clinical validation of a multiplex kit for RAS Mutations in colorectal cancer: Results of the RASKET (RAS KEy Testing) Prospective, Multicenter Study. EBioMedicine 2015, 2, 317-23. doi: 10.1016/j.ebiom.2015.02.007.

・Taniguchi, H.; Okamoto, W.; Muro, K.; Akagi, K.; Hara, H.; Nishina, T.; Kajiwara, T.; Denda, T.; Hironaka, S.; Kudo, T.; et al. Clinical validation of newly developed multiplex kit using Luminex xMAP technology for detecting simultaneous RAS and BRAF mutations in colorectal cancer: Results of the RASKET-B Study. Neoplasia 2018, 20, 1219-1226. doi: 10.1016/j.neo.2018.10.004.

We assess KRAS mutation in the primary colon tumor. A meta-analysis concluded that KRAS status was the same in primary colon tumor and metastatic tumor. (Baas JM, Krens LL, Guchelaar HJ, et al. Concordance of predictive markers for EGFR inhibitors in

primary tumors and metastases in colorectal cancer: a review. Oncologist. 2011;16:1239-49.)

Reviewer #2-3:  In the results section line 119, the patient’s number is 67 not 46.

Reply 2-3:

We appreciate you pointing out our mistake. We revised the number.

Reviewer #2-4:  As KRAS mutations most probably are found in all CRLM performed in the same patient, I would recommend the authors to complete analysis comparing both groups of 67 patients with wKRAS and mKRAS without separating CLRM for each patient.

Reply 2-4:

The aim of this study was to investigate the impact of KRAS status on surgical margin width for local recurrence rates (LRR) after CRLM resection, as mentioned in introduction section, and the primary endpoint was the LRR in each group, as mentioned in materials and method section. In order to observe local recurrence, each tumor must be targeted individually. Local recurrence cannot be an endpoint when the patient, not the tumor, is the subject of the analysis. Our method is based on that of other previous reports, for example, “Procopio F, Viganò L, Cimino M, Donadon M, Del Fabbro D, Torzilli G. Does KRAS mutation status impact the risk of local recurrence after R1 vascular resection for colorectal liver metastasis? An observational cohort study. Eur J Surg Oncol. 2020 May;46(5):818-824. doi: 10.1016/j.ejso.2019.12.004.” Certainly, if the number of tumors per patient is skewed between the KRAS mutation positive and negative groups, the study results could be biased. The number of tumors per patient in both groups was compared and no significant difference was found between the two groups in this study. Thank you.

Reviewer #2-5:  I would recommend showing all three graphs in figure 2 in the same page.

Reply 2-5:

                  We revised three graphs as your suggestion. We will leave the final page layout to your journal's editorial team. Thank you.

Reviewer #2-6: Adjuvant treatment might affect recurrence and was not addressed/discussed in the results section.

Reply 2-6:

It was difficult to include the presence or absence of chemotherapy in the results because the analysis was performed with the local recurrence rate as the endpoint. It was difficult to analyze the groups according to the presence or absence of KRAS mutations, the presence or absence of chemotherapy, and also according to the width of the surgical margin. We agree that chemotherapy is an important factor for local recurrence. But chemotherapy was not the primary endpoint. Further studies focusing on preoperative and postoperative chemotherapy are needed as mentioned in discussion section and limitation sections. “Although, no strong evidence has been reported for the efficacy of preoperative chemotherapy for CRLM, many experts highlight its importance in reducing local and early postoperative recurrences. Furthermore, it is speculated that mKRAS tumors may be resistant to chemotherapy, resulting in residual live tumor cells at the margins, contributing to local recurrence. More than half of the patients received preoperative chemotherapy in the present study and the response was neither radiologically nor pathologically investigated at this time. Thus, further research is needed.” This is our limitation, and another study is needed to analyze this. We performed additional analyses for preoperative and postoperative chemotherapy. In this study, preoperative or postoperative chemotherapy were not risk factors for local recurrence. We revised limitation section as shown below Reply 2-7.

Reviewer #2-7: I would recommend elaborating more on limitations (see comment #2, #4 and #6). In addition, examination of previously operated primary tumor might affect the surgical margin (line 259-261) which might result in bias.

Reply 2-7:

                  We revised the limitation section on line 278 as follows. This study had several limitations. This was a single-center, retrospective study. The sample size was small; particularly, a limited number of samples had wide surgical margin widths. Patients with tumors of varying origins were included in this study. Therefore, large prospective studies are warranted. The aim of this study was to investigate the impact of KRAS status on the surgical margin width for LRR, and the primary endpoint was the LRR. To observe local recurrence, tumors and not patients, should be targeted individually 38. If the number of tumors per patient is skewed between KRAS mutation positive and negative groups, the study results may be biased. The number of tumors per patient in both groups was not significantly different between the two groups in this study. We would like to consider limiting the observation to single liver metastases.

The relationship between preoperative and postoperative chemotherapy and the resection margin was not analyzed in this study and is a topic for future research. Studies from a pathological perspective, such as TBC, may also lead to new findings.

We added the reference associated with above. Procopio, F.; Viganò, L.; Cimino, M.; Donadon, M.; Del Fabbro, D.; Torzilli, G. Does KRAS mutation status impact the risk of local recurrence after R1 vascular resection for colorectal liver metastasis? An observational cohort study. Eur J Surg Oncol 2020, 46, 818-824. doi: 10.1016/j.ejso.2019.12.004.

Your suggestion, “In addition, examination of previously operated primary tumor might affect the surgical margin (line 259-261) which might result in bias.”, this is a misunderstanding. Until the end of this study in January 2023, surgical procedures or surgical margin width targets have never been changed by RAS status of prior primary surgical specimens. The sentence you pointed, “If the case is metachronous CRLM, examination of the previously operated primary tumor will reveal the KRAS status. Hence, if an mKRAS tumor is detected, a surgical margin width of at least 5 mm should be planned. However, for CRLM with wtKRAS, a surgical margin of 1 mm is sufficient for local control, and the surgery may be less difficult and minimally invasive.”, is our “new strategy and proposal” from our result.